# UPF1—From mRNA Degradation to Human Disorders

**DOI:** 10.3390/cells12030419

**Published:** 2023-01-27

**Authors:** Jacek Staszewski, Natalia Lazarewicz, Julia Konczak, Iwona Migdal, Ewa Maciaszczyk-Dziubinska

**Affiliations:** 1Department of Genetics and Cell Physiology, Faculty of Biological Sciences, University of Wroclaw, 50-328 Wroclaw, Poland; 2Institute of Genetics and Development of Rennes, CNRS UMR 6290, University of Rennes 1, 35000 Rennes, France

**Keywords:** UPF1, helicase, E3 ligase, NMD, decay pathways, cancer, neurodegenerative disorders

## Abstract

Up-frameshift protein 1 (UPF1) plays the role of a vital controller for transcripts, ready to react in the event of an incorrect translation mechanism. It is well known as one of the key elements involved in mRNA decay pathways and participates in transcript and protein quality control in several different aspects. Firstly, UPF1 specifically degrades premature termination codon (PTC)-containing products in a nonsense-mediated mRNA decay (NMD)-coupled manner. Additionally, UPF1 can potentially act as an E3 ligase and degrade target proteins independently from mRNA decay pathways. Thus, UPF1 protects cells against the accumulation of misfolded polypeptides. However, this multitasking protein may still hide many of its functions and abilities. In this article, we summarize important discoveries in the context of UPF1, its involvement in various cellular pathways, as well as its structural importance and mutational changes related to the emergence of various pathologies and disease states. Even though the state of knowledge about this protein has significantly increased over the years, there are still many intriguing aspects that remain unresolved.

## 1. Introduction

Up-frameshift protein 1 (UPF1) was firstly described more than four decades ago in yeasts [1] and, since then, we have learned about the variety of functions of this protein. It is known that UPF1 is ubiquitously present and evolutionary conserved among almost all organisms, which could suggest its importance for cellular metabolism from the very early stages of organism development [2,3,4,5]. Orthologs of this protein are highly similar in structure; thus, mutations at specific sites exert comparable effects on the regulation and known functions of UPF1 [6,7,8]. Its first and major role has been described in the context of mRNA decay pathways. Among such processes, the best known and studied is NMD. It has been investigated in several organisms, such as *Saccharomyces cerevisiae*, *Drosophila melanogaster*, *Caenorhabditis elegans* and humans [9,10,11,12,13]. Initially, this process was thought to be the main mechanism for inhibiting the production of damaged proteins. However, over the course of advancing research, it was found to have broader functions for post-transcriptional regulation in both normal physiological conditions and in response to different stresses [14]. Besides the well-known role of UPF1 in the NMD pathway, its involvement in many other RNA degradation pathways has also been investigated, including glucocorticoid-receptor-mediated mRNA decay (GMD), replication-dependent histone mRNA decay (HMD), regnase-1-mediated mRNA decay (RMD), Staufen (STAU)-mediated mRNA decay (SMD) and Tudor-staphylococcal/micrococcal-like nuclease (TSN)-mediated microRNA decay (TumiD) [15]. Besides its importance in RNA turnover pathways, several studies also proposed UPF1 involvement in protein degradation, as it contains a RING-like domain, characteristic of the E3 ligase enzyme group. However, it has not yet been well established whether UPF1 participates as a ubiquitin ligase in cells and controls protein levels of certain substrates or the production of abnormal peptides formed from PTC-containing mRNAs [16,17]. Therefore, understanding all aspects of UPF1′s mechanisms of functioning in cells seems to be substantial, as UPF1 also participates in pathological conditions. Numerous dysfunctions of this protein have been described as an underlying cause of many diseases, and its roles in neurodegeneration and cancerogenesis have been extensively examined in many studies throughout the last decade [18,19]. The upregulation and downregulation of UPF1 has been ascertained in numerous cancers, depending on their type. Additionally, UPF1 was found to be involved in the cellular response towards viral infection [20]. Considering the multitasking characteristic of UPF1, publications that describe novel functions of the UPF1 protein are being published every year. Below, we present some insight into the UPF1 structure and known processes in which UPF1 plays a pivotal role, bearing in mind that, most presumably, this protein still harbors many unknown functions.

## 2. UPF1 Structure

Sequential and structural analyses of yeast and human UPF1s revealed that they are remarkably similar proteins that fold into a characteristic helicase core with several regulatory domains [8,21] (Figure 1). Properties of those domains have been studied to elucidate their importance and regulatory effects on UPF1 role as RNA-dependent ATPase, RNA/DNA helicase and beyond. In particular, the conserved cysteine/histidine-rich domain (CH-domain) is essential for all known functions of UPF1, including its putative ubiquitin ligase activity [16,21,22,23,24,25,26]. Following that, despite the several differences in the triggering mechanism of yeast and human NMDs, phosphorylation and dephosphorylation seem to be of considerable importance for both the proper activation of UPF1 and its further function in aberrant mRNA decay [23,27,28,29,30]. Numerous factors responsible for those processes, e.g., UPF2 and UPF3 (up-frameshift two and three nonsense-mediated mRNA decay factors), the suppressor of morphogenesis in genitalia-1 (SMG1) kinase, PP2A (protein phosphatase 2A), SMG5-7 and the exon junction complex (EJC), have been identified in humans and characterized for a better understanding of UPF1–protein complex associations, remodeling and NMD activation [24,27,31,32,33,34].

Yeast Upf1, otherwise known as the nuclear accommodation of mitochondria (Nam7), is a 109 kDa protein, whereas human UPF1, also described as the regulator of nonsense transcripts 1 (RENT1) or SMG2, is larger and has a mass of 130 kDa [35,36]. As a result of alternative splicing, human UPF1 exists in two isoforms. UPF1_SL_ from the “short loop” consists of 1118 amino acid residues, whereas the “long loop” UPF1_LL_ differs only in the length of a regulatory loop in the 1B domain due to 11 amino acid insertions. Usually, UPF1 refers to the shorter version, which accounts for the majority of UPF1 mRNA [37,38]. Both human and yeast proteins contain a characteristic helicase core structure and main 5′-3′ RNA unwinding activity [6,7,8,21,35,39]. In general, helicases can be divided into several superfamilies (SF). Structural and functional studies divide such families into two groups: those forming toroidal multimeric structures (SF3-6) and two superfamilies of nontoroidal helicases SF1 and SF2 [40]. Based on structural aspects of the helicase core, UPF1 is classified as a member of SF1 [35,40]. Applequist et al. showed that the short isoform sequence of human UPF1 (UPF1_SL_) is highly similar to previously characterized yeast Upf1 protein, with a 51% identity match. The human UPF1 helicase domain, such as yeast Upf1, consists of seven motifs typical to SF1 and SF2 helicases, but also possesses some additional regions, rich in proline/glycine and containing serine/glutamine and serine/glutamine/proline repeats (SQs). Both proteins assume an almost identical monomeric structure with two major domains [8,21] (Figure 1). In proximity to N-terminus lies a zinc-finger region, rich in cysteine and histidine residues (CH-domain), followed by the helicase domain (HD) comprised of previously mentioned seven common ATPase and helicase motifs within two RecA-like folds [8,21] (Figure 1).

### 2.1. Helicase Core Structure

Structural studies revealed that the human UPF1 helicase core domain (residues 295–914) consists of α-helical stalk region and two RecA-like α/β domains, designated as 1A and 2A, with two inserted subdomains 1B and 1C (Figure 1 and Figure 2) [8,21]. Domain 1A (449–555 and 610–700) folds into ten helices, while domain 2A (701–914) forms β-sheet flanked with six helices in total. A nucleotide-binding site is located between those two folds in a characteristic cleft. Cheng et al. suggested that a similarity in RecA-like domain 3D architecture with other helicases may be highly conserved evolutionarily [8]. Subdomains 1B (residues 325–414) and 1C (556–609) form two additional entities, β-barrel ‘above’ the contact of 1A and 2A domains and three helices over 1A, respectively. Seven classical sequence motifs of SF1/SF2 helicases are located in 1A (motifs I, Ia, II and III) and 2A (motifs IV, V and VI) [8,40]. The CH-domain localizes itself ‘above’ the 2A domain, next to 1B (Figure 2) [21].

### 2.2. Nucleotide and RNA Binding

UPF1 possesses RNA-dependent, inducible ATPase activity. Intriguingly, while RNA upregulates the ATPase activity of UPF1, ATP association seems to reduce the affinity of UPF1 to RNA [6,7,34,41,42]. In human UPF1, ATP/ADP binds inside a cleft in a way that the adenine base contacts Y702 and P469 residues, which form hydrophobic pockets (Figure 2A). Q665 is suggested to be responsible for sensing the presence of gamma-phosphate and transducing signals by inflicting conformational changes throughout the molecule. Substitutions of conservative K498, R865 and R703 residues eliminate ATP binding and hydrolysis, while Q665 substitution has no influence on ATP binding, but abolishes ATPase activity. The same was observed upon the mutation of corresponding residues in yeast Upf1 [6,8]. During the ATP hydrolysis cycle, domain 2A moves approximately 20° in relation to 1A due to cleft narrowing or widening after ATP binding or hydrolysis, respectively. These changes influence the positions of the 1B and 1C domains, probably inflicting the catalytic activity of UPF1. Interestingly, molecular and biochemical characterization showed that ATP/ADP binding rather than hydrolysis was altering the UPF1–RNA complex [8]. RNA binds to human UPF1 in a characteristic channel with a 5′ end at the 2A and 3′ end at the 1A side, pointing between domains 1B and 1C, where K599, R600 and R604 of domain 1C form a positive patch. The UPF1 deletion mutant of domain 1C is defective in RNA binding and NMD, thus, indicating the importance of the 1C domain in RNA binding [8]. It is worth noting that the original crystal structure of human UPF1 with RNA was resolved without CH-domain. However, RNA binding in yeast Upf1 with the CH-domain present is comparable, as later studies revealed [21]. Both yeast and human UPF1 structure predictions are vastly similar (Figure 2A,B).

### 2.3. CH-Domain/RING-Like Domain

The CH-domain structure was determined by Kadlec et al. as a unique pseudo-twofold symmetric, antiparallel β-sheet with flanking loops. Within that structure, three zinc atoms occur in different variants of zinc fingers, coordinated by conserved cysteine and histidine residues (Figure 2A), suggesting their importance for the proper structure and function of all UPF1 proteins [22,23]. Interestingly, a surface analysis indicated that conserved residues tend to concentrate and form two potential interaction sites. Indeed, mutational studies confirmed that both surfaces contain residues essential for UPF2 binding [23]. The interaction of the CH-domain with UPF2 was further presented in structural studies, showing that the inherently unstructured C-terminus of UPF2 folds on binding with UPF1, forming β-harpin and α-helix on opposite sides of the CH-domain, corresponding to surfaces formed with conserved residues highlighted in studies of Kadlec et al. [22,23].

It was shown that UPF2 binding stimulates UPF1 ATPase and helicase activity [43]. However, similar results or even greater activation was achieved through the total removal of the CH-domain in UPF1, indicating its inhibitory influence, potentially alleviated with UPF2 binding. Indeed, further structural studies [21] revealed that the CH-domain in yeast Upf1 adheres to the RecA-like 2A domain helicase core and drastically changes its relative localization upon UPF2 binding, translocating itself to the peripheral side of the 1A domain and pointing out of the helicase core structure [21,22]. The docking of the CH-domain to 2A utilizes the same pocket surface as the UPF2 helix, thus, competes for binding. Consistently, the incorporation of the F192E mutation (in human UPF1), which is believed to distort this hydrophobic pocket in the CH-domain, enhances UPF1 ATPase and helicase activity to a similar extent as the deletion of the CH-domain, proving that the dissociation of the CH-domain from the RecA-like 2A domain is required for UPF1 activation [21]. Interestingly, Clerici et al. [22] claimed that the relative localization of the CH-domain, which is in contact with the 1A domain pointing out of the helicase core, is stable with or without the binding of UPF2, and UPF2 binding does not promote the ATPase activity of UPF1. Those discrepancies may be due to other factors involved in UPF1 activation, such as UPF3, or differences in experimental procedures. The CH-domain is also responsible for eRF3 (eukaryotic release factor 3) binding [24] and the interaction with Rsp26 ribosomal protein [25].

As highlighted before, UPF1 is suspected to act as a E3 ubiquitin ligase [16,26]. A The structural analysis of the CH-domain uncovered two RING-like or U-box-like structural modules, similar to E3 ubiquitin ligases (residues 121–172 and 180–233), although this could not be identified based on the sequence itself [23]. Supporting the Upf1 putative E3 ligase activity, Kuroha et al. showed that Upf1 promotes the proteasomal degradation of truncated proteins [17]. Somewhat in agreement with potential E3 ligase activity of Upf1, de Pinto et al. reported that some Upf1 point mutations (C65S, C84S and C148S) localized in the CH-domain are viable in NMD and do not cause respiratory impairment, but do fail to repress mitochondrial splicing deficiency (MSD) while overexpressed. This indicates that the overproduction of Upf1 complements MSD independently from its function in NMD [31,44]. The possible role of Upf1-mediated ubiquitination in protein degradation or NMD in yeast is still under examination. It was shown, however, that human UPF1 is engaged in the proteasome-dependent reduction in the myoblast determination (MYOD) protein level without altering the quantity of MYOD mRNA [16]. Yet, other potential targets for human UPF1 ubiquitination still have not been identified. Intriguingly, human UPF1 was found to be monoubiquitinated by TRIM25 (tripartite motif-containing protein 25) E3 ligase at K592 located in the 1C subdomain, next to the predicted T595 phosphorylation site (residues from the long UPF1_LL_ isoform, an equivalent of residues K581 and T584 from the short UPF1_SL_ isoform) [45]. This indicates that the ubiquitination of UPF1 may have a regulatory function as the phosphorylation/dephosphorylation cycle of UPF1 is necessary for NMD. In yeast, however, the equivalent of T595 is substituted for valine.

### 2.4. Importance of Phosphorylation and Dephosphorylation for UPF1 Function

It was shown that phosphatidylinositol 3-kinase-related protein kinase SMG1 is responsible for human UPF1 phosphorylation, whereas SMG6 and the SMG5–SMG7 complex mediates UPF1 PP2A-dependent dephosphorylation, and that both processes are necessary for NMD [28,29]. The SMG1-dependent phosphorylation of UPF1 certainly occurs on T28 and several serine residues, including S1073, S1078, S1096 and S1116 within the SQ/SQP region, rich in serine/glutamine/proline repeats [28,29,35]. Additional phosphorylation sites were also predicted in a meta-analysis [46,47]. Interestingly, yeast Upf1 does not possess such a SQ C-terminal region [36]. Moreover, no SMG1 ortholog or Upf1-targeting kinase or phosphatase have been identified yet. Only yeast Ebs1 (EST1-like BCY1 suppressor 1) protein, similar in structure to SMG7, has been reported to play a role in NMD [36,48]. Nevertheless, yeast Upf1, as well as Upf2, are also phosphorylated, and such a modification is necessary for NMD [44,49]. In mass spectrometry studies, Lasalde et al. identified eleven new phosphorylation sites in yeast Upf1, of which five (T194, S492, Y738, S748 and Y754) are conserved in human UPF1 and other high eukaryotes. A mutational analysis indicated additional putative phosphorylated residues, Y738 and adjacent Y742, that are crucial for the Upf1 function, but probably in a redundant manner [50].

In human cells, an association of UPF1–SMG1–eRF1/3 (SURF complex) with UPF2–UPF3 proteins from the EJC complex upon PTC recognition is required for UPF1 phosphorylation [27,29]. Interestingly, an alternative SMG1-dependent mechanism of UPF1 phosphorylation, independent of UPF2 binding, was also presented [24]. UPF1 phosphorylation probably triggers NMD complex machinery reorganization, UPF1 5′-3′ helicase activity and mRNA degradation. Hyperphosphorylated UPF1 is a binding target for SMG6 endonuclease (at phosphorylated T28) and SMG5–SMG7 (at phosphorylated S1096 via SMG7). Such an association promotes the dissociation of the complex from the ribosome at the PTC site and helicase activity, and is necessary for PP2A-dependent dephosphorylation, which seems to be required for UPF1 dissociation from mRNA and the recycling of NMD machinery [29,32]. The phosphorylation of UPF1 may also result in the repression of translation initiation through direct eIF3 (eukaryotic initiation factor 3) binding to phospho-UPF1, thus, preventing the joining of the 60S ribosomal subunit to mRNP and the formation of the mature 80S ribosome [30].

### 2.5. eRF1/3 and UPF3 Binding, SURF and Surveillance Complex Formation in UPF1 Activation

Both human UPF1 and yeast Upf1 interact with the eukaryotic release factors eRF1 and eRF3 [34], linking translation termination to mRNA decay. RNA binding with Upf1 does not affect the eRF1 interaction, but reduces the affinity to eRF3, suggesting competition [34]. In agreement with that, ATP binding increases eRF3 affinity to Upf1, as it reduces RNA binding. Thus, succinctly, eRF1/3 binding represses Upf1 RNA-dependent ATPase and helicase activity. Upf1 itself binds the eRF3 independent of GTP and ATP [34,51]. Nevertheless, GTP binding to eRF3 is necessary for the termination of translation and NMD, as GTP is required for the eRF1–eRF3 interaction [51]. Presumably, upon the occurrence of the translation termination event, GTP hydrolysis results in a release of eRF3–Upf1 from the eRF1/3-Upf1 complex, allowing for the Upf2–Upf3 association and formation of the mature surveillance complex and Upf1 activation [24,51,52]. Both yeast and mammalian poly(A)-binding proteins (PABP) interact with eRF3 [53], stimulating translation termination. The eRF3 binding site of human UPF1 corresponds to the PABPC1 binding site, and, indeed, their effect on translation termination is antagonistic in a way that UPF1 inhibits and PABPC1 promotes the termination. This may allow to determine whether the ribosome stall at the stop codon occurs in the natural or premature translation termination site [24]. However, in vitro translation assays showed that human UPF1 has no influence on translation termination [54,55], and that the UPF3B human variant has an actual influence on translation termination, slowing the release of peptides on the PTC site [54,56]. Additionally, a direct interaction between the human UPF3B variant and UPF1 has been revealed, alongside the fact that UPF3B binds with eRF3a [54]. These data indicate that the UPF1 interaction with eRF1/3 is, in fact, indirect. It was also shown that yeast Upf1 directly binds the 40S ribosomal subunit via the Rps26 ribosomal protein, utilizing the CH-domain [25]. Different studies revealed that Upf1 contacts the E site of the 80S ribosome, but this interaction is mediated via the 1C domain. Moreover, Upf1 did not alter the in vitro translation elongation and termination [55]. Taken together, UPF1 probably remains inactive and does not affect translation termination directly.

As mentioned before, the activation of the human UPF1 requires the sequential SURF complex association with UPF2-3 and EJC; thus, resulting in the formation of the surveillance complex and UPF1 phosphorylation [27]. However, another model with distinct branched mechanisms of activation was presented: (I) UPF2/RNPS1-dependent and (II) utilizing BTZ (Barentsz), Y14 (also known as the RNA-binding motif protein 8A (RBM8)), MAGOH (mago nashi homolog), and eIF4A3 of the EJC complex [31]. It is noteworthy that the UPF3B variant is necessary for both NMD pathways, and that they seem to prefer the degradation of different transcript targets [31]. Melero et al. conducted a cryo-EM analysis of the UPF–EJC surveillance complex, which revealed that it forms three structure modules. The central one, consisting of UPF2, the RRM-domain of UPF3 and the UPF1 CH-domain, is flanked by the UPF1 helicase core module on the 3′ side and EJC with the UPF3 C-terminal domain module on the 5′ side. Curiously, UPF1 points to the 3′ side of the EJC complex, which actually stays in agreement with its 5′-3′ helicase activity, but differs from previous models [57]. Further cryo-EM studies elucidated that some part of two RecA-like domains interacts with the C-terminal region of the SMG1 kinase complex (SMG1C), and that, in the vicinity, UPF2 contacts SMG1 independent of UPF1 [33]. Melero et al. suggested that SMG1C can recruit UPF1 and UPF2 separately or in an already established complex, and that UPF1 and UPF2 can interact in a context of the whole complex [33]. Despite the moderately broad knowledge about the UPF1 structure, its role in PTC recognition, translation termination, SURF complex formation, the UPF2/UPF3/EJC interaction and surveillance complex organization, little is still known about the exact remodeling of all the protein complex machineries in transition states of UPF1 activation and the initiation of mRNA decay. Thus, further research is needed to elucidate the process of UPF1-dependent NMD.

## 3. UPF1 Functional Role and Importance

Given the structure and variety of interactions of UPF1 with different proteins, it is not surprising that UPF1 is involved in many cellular processes. It is also worth emphasizing the diversity of these processes, from the regulation of RNA breakdown to the control of protein degradation or the formation of aggresomes (Figure 3).

### 3.1. Nonsense-Mediated Decay Pathway (NMD)

In eukaryotes, one of the evolutionarily conserved pathways and the most important one in controlling and reducing the formation of nonfunctional truncated forms of proteins is NMD. The NMD pathway is a cytoplasmic, translation-dependent process that reduces the half-life of the transcripts due to the presence of PTCs or abnormally long 3′ untranslated regions (UTRs). In yeast, PTC is defined independent of exon boundaries, in contrast to mammalian cells, where PTC-containing transcripts are dependent of their position in the mRNA [58]. During pre-mRNA formation, a multiprotein EJC of approximately 20–24 nucleotides is attached prior to exon–exon fusion. The EJC–mRNA complex is transported into the cytoplasm. Then, during the normal translation at the ribosome, the EJC is removed [59]. The ribosome in the PTC becomes blocked, leaves the EJC downstream and the distance to the 3′ end and the actual STOP codon of the poly(A) tail becomes too large to allow for translation termination [60]. Thus, due to a delay in ribosome release, the recruitment of factors involved in NMD and other cofactors may occur. In the case of the translation of the correct transcript, the stop codon in the last exon removes all EJCs [61]. As mentioned before, among NMDs, the UPF1 protein is one of the core proteins that in human cells is recruited by the eRF1/3 complex together with the SMG1 protein. Together, they create a SURF complex. In such a complex, UPF1 is phosphorylated via the SMG1 protein, which directly inhibits the transcript translation and further recruits mRNA degradation subunits [30]. The triggering of NMD directly dependents on the phosphorylation of UPF1, as well as its recruitment downstream from the termination codon (TC) [62,63]. Following UPF1 phosphorylation, the NMD substrate is successively remodeled. Due to these substrate changes, translation initiation is inhibited, and subsequent mRNA degradation machinery components are recruited [54]. Strikingly, besides the important conservative element, namely, UPF1, in this machinery, the mechanism of NMD action and the recruitment of other elements differs between eukaryotes.

In addition to its role in quality control, the NMD pathway is also responsible for the amount of mRNA, controlling the number of transcripts in the cell. Notably, NMD is presented as a highly efficient process that eliminates aberrant transcripts accounting for up to 30% of total mRNA [64,65,66]. The maintenance of the homeostasis of normal cellular transcripts has also been evolutionarily dependent on the NMD process. It was estimated that NMD regulates the stability of ~5%–10% of normal, physiologic mRNAs [67], and its activity can be modulated in response to developmental changes or environmental stresses. In humans, this pathway has been shown to play a key role in processes such as cell cycle regulation, cell viability or maintaining telomere integrity [68,69]. In the stress response, NMD can employ several strategies to counteract stress and restore homeostasis or, alternatively, lead cells into the apoptotic pathway. For instance, NMD can be repressed in response to stress such as nutrient reduction and hypoxia [70], or reactive oxygen species (ROS) [71]. Recent studies also revealed that human UPF1 isoforms SL and LL may differ in specificity towards RNA targets with a higher affinity for UPF1LL to NMD-resistant transcripts. This effect may result from the reduced sensitivity of UPF1LL to mRNA shielding properties of PTBP1 (polypyrimidine tract-binding protein 1) and may be associated with ER and the translation stress response [38].

### 3.2. Staufen1 (STAU1)-Mediated mRNA Decay (SMD)

Originally, STAU1 was identified as a factor that functions to localize maternal mRNAs in *Drosophila* oocytes and eggs [72,73,74,75,76]. SMD is a mammalian mRNA degradation pathway that is guided by the STAU1 protein. It binds to a STAU1-binding site (SBS) within the 3’ UTR of target mRNAs [74]. Both STAU1 and its paralog STAU2 interact directly with the human UPF1, increasing its helicase activity to promote an effective SMD process. Interestingly, an escalated ratio of STAU1 to UPF2 level results in more SMD while decreasing the amount of NMD events, as both of these proteins compete for the binding site on UPF1 [77]. As in NMD, the enrichment of UPF1 in the 3’ UTR causes UPF1 to interact with the terminating ribosome, typically also for STAU1-mediated mRNA degradation. Markedly, STAU1 proteins are also involved in the localization of mammalian mRNA in neurons and oocytes [77,78]. SMD is known to target a variety of transcripts. Thus, it is also involved in various cellular and physiological processes, such as myogenesis, adipogenesis, cell motility and autophagy [74,79,80,81].

### 3.3. Replication-Dependent Histone mRNA Decay (HMD)

Histone and DNA synthesis are highly regulated processes and require an appropriate balance between the amount of DNA and histone proteins. In most mammals, the synthesis of histones occurs after transcription, at the level of the regulation of the amount of histone mRNA. Histone mRNA levels increase as cells enter the S phase and are rapidly degraded towards the end of this phase. Additionally, the inhibition of DNA synthesis itself causes the rapid degradation of histone mRNAs [82,83]. Histone mRNAs are the only transcripts that are not polyadenylated and, instead, have a poly-A tail end in a conserved stem–loop structure (SL) [84,85,86,87]. The stem–loop starts between 20 and 75 nucleotides before the termination codons. It is specifically bound with stem–loop-binding protein (SLBP), which is required for all steps of histone turnover, and both SL and SLBP mediate the rapid degradation of histone mRNAs. Kaygun and Marzluff determined that the degradation of histone mRNAs required the UPF1 protein, as well as the ATM (ataxia–telangiectasia-mutated) and Rad3-related protein kinases (ATR), which are necessary for the replication stress response on DNA damage [86]. Moreover, it was presented that UPF1 causes histone mRNA uridylation and degradation in response to the DNA synthesis blockage. Such a system was determined and described in *Aspergillus nidulans*, where UPF1 is necessary for the correct regulation of histone mRNAs [87,88,89]. However, there are still many speculations in terms of the mechanism and UPF1 regulation in this process. Human UPF1 is abundant, and is localized between the stem–loop and termination codon regions. It is believed to be activated by SMG1-mediated phosphorylation [90] or by the PI3 K-like kinases ATR and DNA–PK [88,91]. It is not yet clear whether the binding of SLBP to a CH-domain or to the domain of UPF1 helicase directly activates protein helicase and ATPase functions, or whether UPF1 phosphorylation is sufficient to accomplish the decay pathway [86,91].

### 3.4. Structure-Mediated RNA Decay (SRD)

Nontranslated fragments of mRNAs are usually highly structured [92,93] and, in turn, have shorter half-lives [92,93,94]. Structure-mediated RNA decay (SRD) is a process of the selective degradation of mRNAs, as well as circular RNAs with highly structured regions. In humans, this pathway is highly dependent on UPF1’s ability, but, at the same time, independent of the RNA sequence linearization [92,95]. Up till now, the binding of UPF1 to highly structured regions of RNA has been shown to be dependent on its helicase function. Most helicase-dependent 3’ UTRs have a highly structured 3’ UTR (HSU), regardless of their length, and exhibit shorter than average UPF1-dependent half-lives and degradation [95]. Furthermore, a meta-analysis of proteomic and CLIP-seq data showed that Ras GTPase-activating proteins interact with UPF1 depending on its ATPase function and associated protein G3BP1 by preferentially binding at the HSU site [95].

### 3.5. Regnase-1-Mediated mRNA Decay (RMD)

In terms of the mechanism, RMD depends on translation termination, similar to NMD and SMD. It is a type of mRNA degradation process that recognizes stem–loop structures in the 3’ UTR with the regnase-1 protein playing a key role in this pathway. It is an endoribonuclease that directly digests mRNA and is involved, among others, in the activation of immune cells [96,97]. The regnase-1 protein has been identified in macrophages and its gene is induced through the stimulation of lipopolysaccharides (LPS and TLR4 (Toll-like receptor 4) ligands) [96,98]. Structural studies of regnase-1 have shown that the PIN domain of the protein has the ability to bind the catalytic site of RNase [99,100]. Further, it was presented in these studies that regnase-1 preferentially digests single-stranded RNAs (ssRNAs) rather than structured ones. In RMD, 3′ UTR-bound regnase-1 works equivalently to the 3′ UTR features from the NMD and SMD pathways. It has not yet been clearly established whether the human UPF1 is recruited either through the SURF complex or through fusion with the 3’ UTR. In the terminal site of the ribosome, UPF1 is part of the regnase-1 termination and translation complex. This can activate the endoribonuclease capacity of regnase-1, as well as the helicase activity of the UPF1 protein, which determines the activity of RMD. However, it remains to be elucidated whether UPF1 and other factors of the RMD complex must be phosphorylated [101,102].

### 3.6. Glucocorticoid-Receptor-Mediated mRNA Decay (GMD)

The glucocorticoid receptor (GR) functions as a hormone-dependent transcription factor that regulates diverse biological processes, such as stress response pathways or inflammatory reaction [103]. It was first studied in terms of its function as a nuclear SMG1C receptor; however, its function in the RNA degradation process was only known because GR can directly bind to RNA. Now, it is clear that, in the GMD process, it participates in a two-step pathway of RNA decay. Noticeably, this process is independent from translation and requires UPF1 recruitment to a 5′ UTR region on natural substrate mRNAs [104,105]. The GR binds to the GC-rich sequences, which can form two loops in the 5′ UTR of the mRNA, leading to the breakdown of bound mRNAs. Importantly, this process is dependent on the GC ligand for the DNA binding, in case the GR binding to RNAs is independent from this substrate. On target mRNA, the GC–GR complex recruits proline-rich nuclear receptor coactivator 2 (PNRC2), which seems to be able to enroll UPF1 and decap mRNA 1A (DCP1A) proteins [104,106].

### 3.7. Tudor-Staphylococcal/Micrococcal-Like Nuclease (TSN)-Mediated MicroRNA Decay (TumiD)

MicroRNAs (miRNAs) are short noncoding RNAs of ∼22 nt. Their main role involves the mediation of gene silencing through a miRNA-induced silencing complex (miRISC). This machinery is conducted to miRNA-binding sites in the 3′ UTR sites of substrate mRNAs [107,108]. There is no information on the degradation pathway of miRNAs, but it is known that they regulate the vast majority of transcripts encoding proteins. The Tudor-staphylococcal/micrococcal-like (TSN) nuclease promotes the decay of miRNAs containing CA and/or UA dinucleotides, described as a TSN-mediated miRNA decay pathway (TumiD). It was reported in humans that cellular TumiD requires UPF1 protein. TNS and UPF1 were shown to directly interact in vitro and form a complex with the RISC components Argonaute RISC catalytic component 2 (AGO2) and GW182. This may indicate that the originally formed TSN–UPF1 complex may transiently interact with RISC-associated miRNA targets [109]. UPF1 can separate miRNAs from their target mRNAs, which makes the miRNAs liable to TumiD. Other functional nonprotein-coding RNA particles, long noncoding RNAs (lncRNAs), which are more than 200 nucleotides in length, also interact with UPF1 [110]. For instance, in recent studies, the long noncoding RNA (lncRNA) CALA was shown to regulate RNA turnover in endothelial cells, affecting the NMD via the G3BP1-RNPs formation containing various NMD factors, including UPF1 [111]. The role of the miRNA and lncRNA interplay with UPF1 in various types of cancer is pondered below.

### 3.8. Viral Targeting of Upf1

Upon the increased level of UPF1 protein that binds to GC-rich highly structured 3′ UTR regions, it is more likely that UPF1 is phosphorylated and interacts with a terminating ribosome, which causes the mRNA decay initiation [112,113]. Since the genome of viruses has a multicistronic structure, it tends to be recognized by the NMD system. Importantly, the 3′ UTR viral structures are GC-rich and have structures that have the potential to recruit the host factors [114]. Various types of viruses become targets for the host’s NMD pathway, such as the retrovirus *Rous sarcoma virus* (RSV), *Semliki forest virus* (SFV), flaviviruses *Hepatitis C virus* (HCV) and *Zika virus* (ZIKV). Strikingly, the DNA virus *Kaposi’s sarcoma-associated herpesvirus* (KSHV) is also targeted by Upf1 and NMD during pre-mRNA splicing [20,115,116]. In this pathway, the role of UPF1 as one of the major regulators of the NMD pathway appears to be fundamental. Apart from NMD, the role of UPF1 in additional RNA degradation pathways in virus metabolism is not examined well. Up till now, investigations found that NMD is drastically inhibited upon a viral attack on plant and animal cells contributing to pathogenesis. Targeting viruses to UPF1 protein seems to be necessary, because all of them have to translate their genomes or antigenomes. For this reason, future work should most likely identify additional virus groups that may undergo decay via the UPF1 protein. The role of UPF1 in future work should also be investigated in terms of host gene dysregulation within NMD disorders and other RNA decay pathway processes following viral infection [20]. The significance of UPF1 in human viral infections is discussed in detail below.

### 3.9. Ubiquitin Ligase Activity

Besides the clear involvement of Upf1 in mRNA decay pathways, some research declares that UPF1 has a function in addition to that, and is involved in the degradation of peptides [16,17,117]. However, there is still a lot of speculation and confusion on this issue. In yeasts, it was observed in several studies that Upf1 led to the degradation of both PTC-containing mRNAs as well as peptides. This depended on the 3′ UTR, as well as on proteasome functionality, and may further suggest that Upf1 can potentially participate on different levels of degradation machinery on both mRNA and peptide quality control [17,117]. It was further studied that Upf1 may interact with cell division cycle 34 (Cdc34), as it was presented with coimmunoprecipitation, and, possibly, this E2 protein may influence its autoubiquitination. Takahashi et al. also presented that the RING-like domain interacts with the Upf3 protein, which enriches the NMD ability. Notably, a Upf2-independent interaction with Upf3 seemed to be required for this self-ubiquitination, but the CH-domain itself failed to interact with Upf3 alone. Direct Upf1–Upf3 contact sites have not been discovered yet, although the coprecipitation of Upf1 with Upf3 was reported on in multiple studies [26,31,32,54]. The only described protein substrate of the UPF1 ligase was presented by Feng et al. in mammalian cells. As it was mentioned, the human MYOD protein level is stabilized upon the UPF1 ligase’s ability deactivation, while its mRNA level remains stable. An examination of lysates from cells expressing a mutated version of UPF1 (S124A/N138A/T139A—from a RING-like fold within the CH-domain), which is viable in NMD but defective to some extent in CDC34 (human ortholog of Cdc34) binding, revealed, inter alia, an impairment in the ubiquitin transfer to MYOD and the absence of polyubiquitinated MYOD. Mutated UPF1 also failed to reduce the level of MYOD in comparison to the wild type, and did not slow the process of myogenesis. It is worth mentioning that the exact influence of such alanine substitutions on the whole CH-domain structure and its regulative function, not only in NMD, is unknown. Additionally, the indirect impact of known UPF1 activity on MYOD ubiquitination and degradation cannot be ruled out. However, data presented by Feng et al. indeed indicated that the proteasomal degradation of MYOD and the suppression of differentiation requires the RING-like domain-dependent E3 ubiquitin ligase activity of UPF1 [16]. Thus, UPF1 contributes to the rapid degradation of normally occurring, well-folded and functionally active proteins, as well as abnormal PTC polypeptides.

### 3.10. Aggresome Formation

As PTC-containing mRNAs produce a shortened protein version upon the disturbance of the ubiquitin–proteasome system (UPS), these products can localize in aggresomes. Indeed, it was investigated in recent studies that upon UPS deactivation, the PTC products were enriched in aggresomes, confirming the important role of the human UPF1 in aggresome formation [118]. Besides PTC products, it seems that UPF1 is also involved in non-PTC protein regulatory machinery. This was examined in the context of mutation in the cystic fibrosis transmembrane conductance regulator [119,120,121], as well as puromycin-conjugated polypeptides [122]. Interestingly, UPF1 involvement in aggresome formation directly relates to its phosphorylation [118,123]. When UPS is overloaded or polypeptides arising from NMD substrates are not adequately ubiquitinated, NMD polypeptides may undergo an additional aggresome-mediated degradation pathway. These products are then selectively recognized with eukaryotic translation elongation factor 1 alpha 1 eEF1A1 [121,124]. This uptake of NMD polypeptides is promoted by the hyperphosphorylation of UPF1 with SMG1 during NMD [125]. In the next step, the complex composition of the nuclear cap-binding complex-dependent translation initiation factor **C**TIF, **e**EF1A1 and dynactin 1 **D**CTN1, namely, CED, with bound NMD polypeptides, moves towards the aggresome via retrograde transport through microtubules [126,127]. Ultimately, NMD products that accumulated in the perinuclear aggresome compartment are eliminated via the autophagy pathway [118]. At the same time, the E3 ligase ability is not necessarily needed for the human UPF1 in aggresome formation [118,122]. Given that UPF1 acts as an enhancer of aggresome formation, the above observations suggested that UPF1 may regulate various cellular and biological processes more than we have considered before.

## 4. UPF1 in Human Disorders

UPF1 plays a substantial role in various human diseases. Its aberrant forms that lead to the dysregulation of NMD were ascertained in numerous types of cancer [19], as well as neurodevelopmental and degenerative disorders [18,128]. UPF1-mediated RNA decay pathways, such as NMD, SMD and SRD, target viral RNAs yielding a decreased efficiency of infection [20]. There are also working antiprion yeast systems that depend on the UPF1 function, with potential applications in human prion infections [129]. Table 1 comprises selected diseases, associated aberrations of UPF1 and other factors, as well as the consequences of those defects.

### 4.1. UPF1 in Cancer

UPF1 is downregulated in numerous cancers, which correlates with poor prognosis and low overall survival (OS) rates. Its low expression causes the dysfunction of NMD, increased levels of toxic transcripts and, consecutively, tumor initiation and progression [19,130,131,133,136,138,139,140,141,142,143,146,147]. Additionally, UPF1 acts in various signaling pathways and promotes undifferentiated stem cell phenotypes, proliferation and metastases. Along with different levels of expression, epigenetic and genetic alterations, as well as the aberrant splicing of UPF1 [130,147], have been demonstrated. Epigenetic alterations, such as the hypermethylation of promoter regions of tumor suppressor genes, lead to the downregulation of their expression and influence tumorigenesis [158,159]. The UPF1 putative promoter region possesses an enriched CpG island in gastric cancer (GC), which was proved to be hypermethylated [136]. Hypermethylation was also detected in hepatocellular cancer (HCC). Treatment with the DNA-demethylating drug 5-Aza-2′-deoxycytidine increases UPF1 mRNA and protein levels in both GC and HCC [136].

As transcription regulation is disrupted in cancer cells, the accumulation of aberrant transcripts has been observed in tumors [160], detected additionally in such high levels due to the downregulation of the NMD factor, most importantly UPF1, in various cancer types [19]. In cell lines derived from non-small cell lung cancer (NSCLC), such aberrant splicing isoforms have been identified as potential templates for producing neoantigens, as some of those forms were proved to be translated into several peptides [143]. Clinical lung cancer specimens were also analyzed for the search of aberrant forms and neoantigens in cancer cells in vivo, with each examined patient possessing such isoforms. A total of 2021 novel isoforms were identified. Such a large number has been proposed to be the result of impaired NMD, as ~30% of those isoforms contained PTCs that should be targeted by UPF1 [143]. An impairment in splicing can also be due to mutations in splicing-related factors, such as U2AF1 (the U2 auxiliary factor 35 kDa subunit) and SF3B1 (splicing factor 3B subunit 1), frequently found in several types of solid tumors. U2AF1 and SF3B1 are the core components of spliceosomes that have been proposed as novel therapeutic targets for cancer [161].

Numerous somatic mutations of the *UPF1* gene have been ascertained in pancreatic adenosquamous carcinoma (ASC) and stated as the first known unique molecular markers of ASC [147]. The perturbation of NMD resulting from those tumor-specific mutations significantly increases the number of aberrant mRNAs that should be targeted by UPF1-mediated NMD. Notably, other NMD factors, namely, *UPF2, UPF3A* and *UPF3B* genes, did not display any detectable mutations in analyzed ASC samples. The point mutations gathered in two regions, one embracing exons 10 and 11 and intron 10 in RNA the helicase domain, and a second one consisting of exons 21–23 coding the SQ domain and the ST-Q motif and introns 21 and 22. These mutations are equally distributed among exons and introns, triggering the alternative splicing of *UPF1* pre-mRNAs by disrupting intrinsic splicing enhancers (ISEs) and exonic splicing enhancers (ESEs) [147]. Those unique mutations are beneficial for ASC diagnosis and create the possibility for NMD substrate-targeted therapies.

The epithelial–mesenchymal transition (EMT) contributes to tumor metastasis, and is regulated by various signaling pathways [132,162]. One of the most important is the TGF-β (transforming growth factor beta) signaling pathway, which induces the EMT through activating Smad signaling [163,164]. Overexpressed UPF1 inhibits TGF-β signaling component genes, *MIXL1* and *SOX17*. Upregulated UPF1 decreases the expression of Smad2/3 proteins, which, in turn, leads to the inhibition of TGF-β signaling [145]. Furthermore, UPF1 alters Smad2/3 phosphorylation, which is required for signal transduction [141]. In colorectal cancer (CRC) tissues and cell lines, UPF1 has been found to be significantly upregulated and exhibit a positive correlation with lymph node metastasis and shorter survival, thus, acting as an oncogene. UPF1 knockout (KO) in colorectal cell lines increases the number of cells in the S phase, therefore, UPF1 promotes cell cycle progression. Furthermore, UPF1 KO promotes apoptosis via increasing DNA damage and inhibiting cell migration, invasion and EMT, as it leads to a higher expression of the epithelial marker E-cadherin and decreased levels of the mesenchymal marker vimentin [132]. UPF1 in CRC can act as a promising diagnostic marker and target for novel therapies.

The upregulation of UPF1 in CRC also leads to chemoresistance in vivo and in vitro to oxaliplatin, a third-generation platinum coordination complex, used for treatment in several types of cancer. Chemoresistance results from the SMG1-dependent phosphorylation of the human topoisomerase II-α (TOP2A) and the maintenance of cell stemness [131]. TOP2A organizes the genome structure, promotes chromosome segregation and is overexpressed in multiple tumors, leading to aggressive phenotypes of the disease and poor prognosis [164,165]. Post-translational modifications, such as phosphorylation, ubiquitination and SUMOylation, regulate TOP2A activity [166]. SMG1 directly phosphorylates UPF1 and possibly induces TOP2A phosphorylation through UPF1. Moreover, UPF1 enhances the stem phenotype of CRC cells in a TOP2A-dependent manner. The underlying mechanism of oxaliplatin chemoresistance possibly arises from the attenuation of DNA damage resulting from TOP2A, induced by oxaliplatin as the phosphorylation of TOP2A increases its enzymatic activity. TOP2A was also proved to be upregulated in CRC. Notably, this chemoresistance was subverted with TOP2A silencing [131]. These results pointed out new possible targets for decreasing oxaliplatin chemoresistance in CRC patients.

LncRNA and microRNA can interact with UPF1, resulting in its tumor-suppressive functions in various cancers [110]. In HCC, UPF1 interacts with lncRNA SNHG6 (small nucleolar RNA host gene 6) and suppresses cell proliferation and migration through inhibiting the TGF-β/Smad pathway. SNHG6 represses Smad7 expression and, in turn, induces Smad2/3 phosphorylation [139,141]. Moreover, in CRC, SNHG6 KO led to decreased UPF1 and p-Smad2/3 levels [133]. Another lncRNA engaged in the TGF-β/Smad pathway is SNAI3-AS1, which mediates cell invasion. After the direct interaction of SNAI3-AS1 with UPF1, the tumorigenesis of HCC is suppressed. Additionally, SNAI3-AS1 KO significantly decreases the levels of p-Smad2/3 [167]. Upregulated in HCC miR-1468, which promotes cell proliferation and colony formation, targets UPF1, leading to its downregulation in HCC. The plasmacytoma variant translocation 1 (PVT1) lncRNA upregulated in breast cancer (BC) has been proposed to act as an oncogene through binding miR-128-3p and UPF1 and promoting EMT and, thus, proliferation and metastasis [130]. Sponging miR-128-3p via competitive binding with PVT1 leads to the upregulation of FOXQ1, which is responsible for inducing EMT through e-cadherin repression [168]. Additionally, miR-128-3p was found to be downregulated in BC [130]. Therefore, PVT1 can serve as a potential therapeutic target in BC. In GC, where UPF1 expression is downregulated due to promoter hypermethylation, a negative correlation between the lncRNA MALAT 1 (metastasis-associated lung adenocarcinoma transcript 1) has been observed. The UPF1 inhibition of gastric cancer progression was reduced by high levels of MALAT1, demonstrating a promising target for gastric cancer treatment [136]. Moreover, the miR-seq data show that the degradation of nearly 50% of potential TumiD substrates in human T24 bladder cancer cells was enhanced through UPF1. UPF1 causes the dissociation of miRNA from their mRNA targets, making them vulnerable to TumiD. One example of targeted miRNA is miR-31-5p, which has been correlated with aggressive tumor phenotypes, cell invasion and poor prognosis in breast and bladder cancer. In the case of BC, miR-31-5p is epigenetically silenced, while, in bladder cancer, TumiD plays a regulatory role in determining the miR-31-5p levels. By reducing miRNA amounts, oncogenic genes can be upregulated, which, in turn, leads to tumorigenesis and metastasis [109].

### 4.2. UPF1 in Neurological Disorders

NMD targets are expressed throughout the brain, with more than 90% identified in the cerebral cortex, which is responsible for cognitive functions, such as attention, memory and overall consciousness. Many of those targets are significantly increased by UPF1 knockdown (KD), and are mutated or misregulated in neuronal diseases, such as spastic paraparesis, amyotrophic lateral sclerosis (ALS), frontotemporal dementia (FTD) and autism spectrum disorder [128]. Recently, the fragile X mental retardation protein FMRP, of which the loss of function is the main cause of intellectual disability and autism spectrum disorders in fragile X syndrome (FXS) [169,170], has been identified as a direct NMD-activated phosphorylated UPF1 interactor, influencing its activity by acting as its repressor. That interplay led to the conclusion that the downregulation of substantial neuronal mRNA in FXS is caused by the stimulation of NMD through FMRP loss [128]. The neuroprotective function through RNA binding and helicase activity of UPF1 independent of the NMD pathway has been shown in several in vitro and in vivo models of ALS [18]. Overexpressed UPF1 mitigates the neurotoxicity of a G_4_C_2_ hexanucleotide repeat expansion in the *C9orf72* gene, ascertained to be the most common factor inducing familial and sporadic ALS and FTD [150,151].

Spinal muscular atrophy (SMA) is a neurodegenerative disorder caused by mutations in the *SMN1* (*survival motor neuron 1*) gene, affecting splicing and leading to the production of PTC harboring less stable transcripts extensively targeted by NMD, aggravating the disease phenotype [152,153]. Some truncated proteins, despite being able to partially retain their functionality, are degraded by NMD, which causes haploinsufficiency [64].

Moreover, the link between NMD and epileptogenesis has been examined [154]. Mooney et al. investigated UPF1 and its phosphorylated form, as well as UPF2 and UPF3B levels in a mouse hippocampus after status epilepticus. They described an increase in UPF1, phosphorylated UPF1 and UPF2 in their model, and a mainly neuronal distribution of UPF1. They also ascertained higher levels of UPF1 in human hippocampi from patients with temporal lobe epilepsy (TLE). As miR-128, which targets the NMD system, including UPF1, is decreased in human epilepsy [171], it can be the cause of increased UPF1 levels. Additionally, the increased binding of UPF1 to the 3′ UTR regions of transcripts in mice has been presented. Mice treated with an NMD inhibitor, NMDI14 [172], had less spontaneous seizures and lower daily seizure rates. All those results suggested an elevated NMD associated with status epilepticus in the hippocampus, leading to epilepsy emergence and progression. The NMD system could possibly act as a target for seizure prevention.

### 4.3. UPF1 in Viral Infections

As stated above, UPF1 plays an essential role in viral infection development. In 2022, Fang et al. analyzed the interactome of the Ebola virus (EBOV) polymerase, and found that UPF1, collectively with another mRNA decay factor, GSPT1 (G1 to S phase transition protein 1 homolog), interacts with EBOV polymerase to promote viral replication. At the onset of the infection, UPF1 leads to a reduction in vRNA and mRNAs levels and fewer cells that are infected, while GSPT1 KD decreased the vRNA level but increased mRNA levels. In later infection, UPF1 and GSPT1 are hijacked, and promote viral replication [155]. In a similar manner, *human immunodeficiency virus* (HIV) exploits UPF1 to increase its infectivity. The depletion of UPF1 through siRNA reduces the infectivity of HIV virions by altering their reverse transcription. UPF1 is crucial for virus assembly and its function seems to be independent from NMD [156]. In the future, mutants of UPF1 with impaired ATP-binding or hydrolysis activity could serve as inhibitors of HIV virion infectivity. Additionally, genomes of viruses can also be targets to alternative UPF1-mediated SMD and SRD pathways [20].

### 4.4. UPF1 in Antiprion Systems

Prions are differed, infectious forms of native proteins. They were first discovered in sheep, where they cause fatal, transmissible encephalopathy named scrapie [173]. In the human infectious form of PrP, protein causes a neurodegenerative disorder called Creutzfeldt–Jakob disease and its variants by forming insoluble amyloid plaques [174]. Yeast possesses numerous variants of single-prion proteins [175], which, in their native form, function in nitrogen metabolism (Ure2), translation termination (Sup35) and provide better adaptation to environmental conditions by facilitating amyloid formation (Rnq1). The infectious forms created by those proteins are called (URE3), (PSI+) and (RNQ+), respectively [176,177]. Yeasts are an easy, cheap and safe model to study prions, creating a good alternative to animal models that sometimes can raise ethical questions. Parallelly, yeast has numerous antiprion systems, reducing amyloid formation and curing variants that do form prions. Son and Wickner discovered the link between Upf1 and (PSI+) levels with general screening for antiprion proteins. In the absence of either Upf1, Upf2 or Upf3, the generation of (PSI+) was elevated 10- to 15-fold. A direct interaction was proved, and the mechanism by which prions are cured was proposed. Upf1 monomers compete with the amyloid fibers of Sup35 or bind to the filament ends, preventing fiber growth [157]. NMD is conserved from yeasts to humans, and detailed analyses of the analogs of homologous systems can be applied to the treatment of neurodegenerative prion disorders in humans by enhancing native interactions [129,157]. Additionally, those studies suggest rather prion- or even variant-specific interactions of factors engaged in antiprion systems.

## 5. Conclusions

Up-frameshift protein 1 (UPF1) is an ATP-dependent RNA helicase [6,39] that has been investigated in a wide variety of cellular processes. Except for its 5′-3′ unwinding activity, UPF1 has been associated with many pathways of RNA decay (such as NMD, SMD, HMD, etc.), but also protein quality control [16,17,26]. Alongside many other factors, it forms several different complexes (SURF, surveillance, etc.), impacting RNA stability, but also modulating translation and protein degradation [15,16,17,24,26,27,30,31,32,34,49,57]. What is noteworthy, but often overlooked, is the unique and conserved structure of UPF1, which is associated with the mechanisms of action and complex interplay with other proteins. Especially crucial for many interactions and the overall role of UPF1 in RNA decay and beyond is the CH-domain [16,21,22,23,24,25,26]. It is worth emphasizing that the recently proposed direct involvement of UPF1 in protein degradation and quality control has not yet been fully understood. In particular, the function of UPF1 as ubiquitin ligase and its mechanism remain vastly unclear [16,17,26].

Apart from all known functions, UPF1 probably still hides its multitasking potential, which needs to be revealed. Few studies published in recent years demonstrated novel activities and processes depending on UPF1 activity [178,179]. The selective profiling of the yeast Upf1:ribosome association led to determining a unique ribosome state forming abnormal ribosome-protected mRNA fragments of 37–43 nucleotides in length, dependent on Upf1′s ATPase activity and not full NMD pathway [178]. Additionally, Upf1′s surveillance function precedes NMD as it can bind to ribosomes translating any mRNA, not only those that need to be degraded, and whose recognition by Upf1 triggers NMD. It has been shown that the mutant form of Upf1 can interfere with normal translation termination and ribosome release, and the results strongly support the existence of at least two distinct functional Upf1 complexes in the NMD pathway [178]. The interaction of UPF1 with YTHDF (the YTH domain-containing family) was also shown, indicating the influence of UPF1 on the degradation of modified transcripts containing N6-methyladenosine (m6A) residues [38]. The role of UPF1 in regulating satellite cell migration and adhesion was also identified, demonstrating the regulatory mechanism of UPF1 in the repair of damaged muscle [179]. UPF1’s function in selected stress conditions was also highlighted [38,70,71], but its actual importance in various other stresses, especially those impacting protein production and degradation, is still to be elucidated.

Lastly, UPF1 is associated with numerous human diseases, such as various types of cancer, where it functions mostly as a tumor suppressor, but its oncogenic character has also been proved [19,134]. Additionally, its roles in numerous neurodegenerative disorders, such as ALS and FTD [18], as well as RNA and DNA virus infections, where it impacts virion replication and participates in viral genome degradation, have been reported on [20]. UPF1 genomic mutations or alterations in expression levels can be used as markers of diseases, and the protein itself serves as a promising therapeutic target.

## Figures and Tables

**Figure 1 cells-12-00419-f001:**
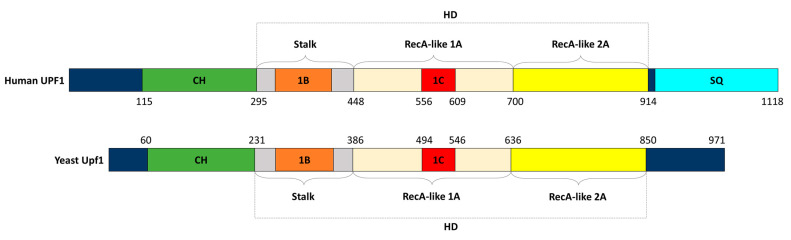
Domain structure of human UPF1 and yeast Upf1 proteins. Cysteine–histidine-rich domain (CH) marked as a green box. Helicase domain (HD) bound with a dashed line box containing stalk region as gray, RecA-like 1A as wheat, RecA-like 2A as yellow and 1B and 1C subdomains as orange and red boxes, respectively. Region rich in serine/glutamine/proline repeats (SQs) marked as cyan box.

**Figure 2 cells-12-00419-f002:**
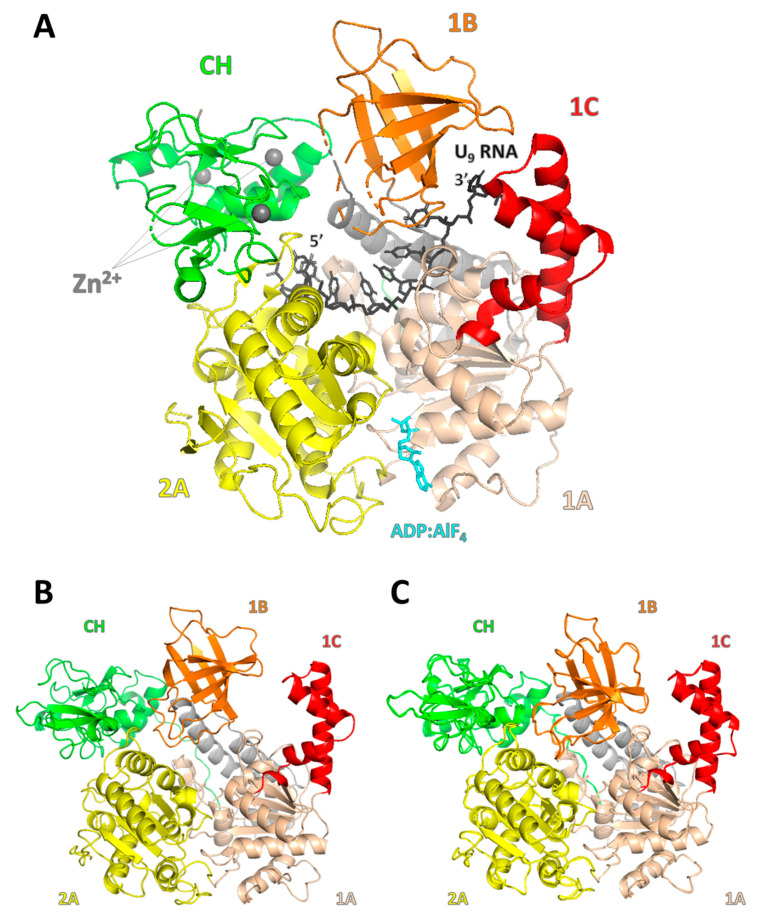
UPF1 protein helicase core structures. (**A**) Yeast Upf1 helicase core crystal structure in Upf1–RNA–ADP:AlF**_4_** complex (residues 61–850, structure accession code 2XZL—protein databank, [21]). RecA-like 1A domain in wheat, RecA-like 2A domain in yellow, CH-domain in green and regulatory subdomains 1B and 1C in orange and red, respectively. Represented as sticks: 9 bp long uracil-RNA (U_9_) in black and ADP:AlF_4_ (mimicking the transition state of the nucleotide in the ATPase cycle) in cyan. Zinc ions within the CH-domain structure are shown as gray spheres. AlphaFold predictions of (**B**) yeast Upf1 (residues 61–850) and (**C**) human UPF1 helicase core structure (295–914). Domains colored as mentioned above. All structure images were created in PyMOL.

**Figure 3 cells-12-00419-f003:**
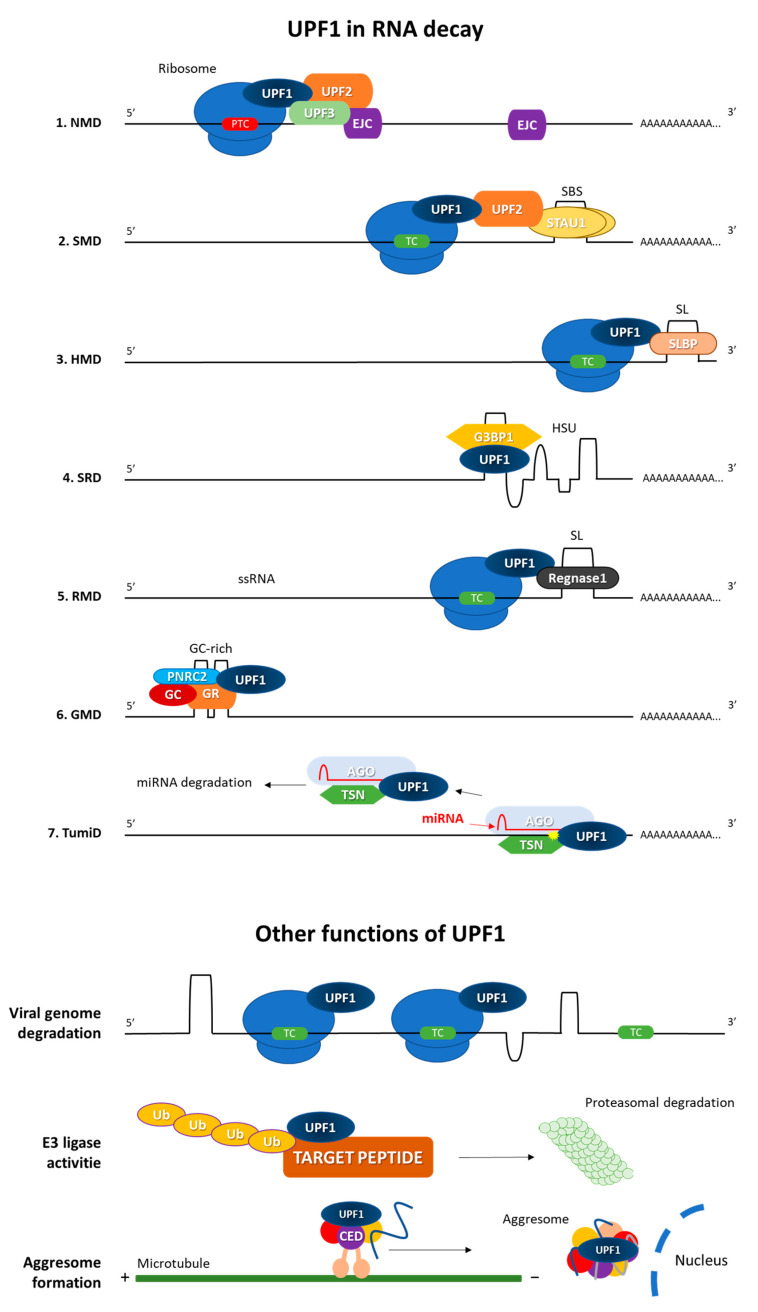
Selected functions of UPF1 protein: pathways of UPF1-dependent RNA decay (1–7) and UPF1 role in viral genome degradation, ubiquitination of peptides and aggresome formation. In RNA degradation, UPF1 is recruited by: (1) translation termination event on premature stop codon (PTC) in proximity to exon–junction complex (EJC) in nonsense-mediated RNA decay (NMD); (2) STAU1 dimer protein associated with STAU1-binding site (SBS) within the 3’-UTR in Staufen1 (STAU1)-mediated mRNA decay (SMD); (3) stem–loop-binding protein (SLBP) associated with a stem–loop structure (SL) in replication-dependent histone mRNA decay (HMD); (4) highly structured 3’ UTR region (HSU) alongside G3BP1 (Ras GTPase-activating protein-binding protein 1) in structure-mediated RNA decay (SRD); (5) regnase-1 protein associated with stem–loop (SL) of mostly ssRNA in regnase-1-mediated mRNA decay (RMD); (6) GR–GC–PNRC2 complex associated with 5′ UTR GC-rich region in glucocorticoid-receptor-mediated mRNA decay (GMD); (7) TSN–AGO–miRNA complex in Tudor-staphylococcal/micrococcal-like nuclease (TSN)-mediated microRNA decay (TumiD). In viral genome degradation, UPF1 is recruited due to DNA/RNA polycistronic (multiple stop codons read as PTC) and a highly structured nature. Additionally, UPF1 may act as E3 ubiquitin ligase-targeting truncated peptides and proteins for proteasomal degradation and is involved alongside CED complex in aggresome formation of NDM-related peptides.

**Table 1 cells-12-00419-t001:** Role of UPF1 in various human diseases. Abbreviations: EMT—epithelial–mesenchymal transition; lncRNA—long noncoding RNA; PVT1—plasmacytoma variant translocation 1; ATF3—activating transcription factor 3; TumiD—Tudor-staphylococcal/micrococcal-like nuclease (TSN)-mediated miRNA decay; RISC—RNA-induced silencing complex.

Disease	Aberration	Effect	References
Cancer Types			
**bladder cancer**	methylation of RISC components, leading to increased UPF1 binding	augmented TumiD, upregulated expression of proinvasive proteins and G1-to-S-phase transition	[109]
**breast cancer (BC)**	UPF1 downregulation by binding with lncRNA PVT1	EMT, proliferation and metastasis	[130]
**colorectal cancer (CRC)**	UPF1 downregulation (microsatellite instable (MSI) CRC)/upregulation (microsatellite stable (MSS) CRC)	EMT, stemness maintenance and oxaliplatin chemoresistance	[131,132,133,134]
**endometrial cancer (EC)**	UPF1 upregulation	stem cell phenotype, metastasis, relapse, chemoresistance and interaction with lncRNA LINC00963 and miRNA miR-508-5p	[135]
**gastric cancer (GC)**	UPF1 downregulation and promoter hypermethylation	proliferation, cell cycle progression and interactions with lncRNA MALAT1	[136]
**glioblastoma multiforme (GBM)**	elevated UPF1 transcriptional levels by ATF3	malignant phenotype, cell stemness and self-renewal	[137]
**glioma**	UPF1 downregulation by binding with lncRNA PVT1	tumor progression and proliferation	[138]
**hepatocellular carcinoma (HCC)**	UPF1 downregulation and promoter hypermethylation	lower interaction with suppressive lncRNAs—UCA1; SNHG6, migration, proliferation and EMT	[139,140,141]
**inflammatory myofibroblastic tumor (IMT)**	UPF1 downregulation, somatic mutations and aberrant splicing	NMD downregulation, immune infiltration, elevated chemokines and IgE levels—IMT characteristics	[142]
**nonsmall cell lung cancer (NSCLC)**	UPF1 downregulation and splice site mutations	neoantigenic aberrant splicing isoforms of proteins	[143]
**lung adenocarcinoma (LUAD)**	UPF1 downregulation/upregulation	EMT, proliferation, invasion and interactions with lncRNA ZFPM2-AS1	[144,145]
**ovarian cancer (OC)**	UPF1 downregulation by binding with lncRNA DANCR	metastasis, proliferation and migration	[146]
**pancreatic adenosquamous carcinoma (PASC)**	UPF1 downregulation, genomic point mutations and aberrant splicing	disruption of exonic and intronic splicing enhancers and NMD target accumulation	[147]
**pancreatic ductal adenocarcinoma (PDAC)**	UPF1 mRNA editing	elevated asparagine synthetase (NMD target) and tumor growth caused by asparagine uptake	[148]
**prostate cancer**	UPF1 cytoplasmic localization instead of nuclear	progression, metastasis, proliferation, cell growth and interactions with plakophilins (PKP) 1 and 3 (cell–cell contacts)	[149]
**Neurological Disorders**			
**fragile X syndrome (FXS)**	UPF1 upregulation through loss of its repressor—fragile X mental retardation protein (FMRP)	FXS phenotype, intellectual disability and autism spectrum disorders, NMD misregulation and molecular abnormalities	[128]
**amyotrophic lateral sclerosis (ALS) and frontotemporal dementia (FTD)**	- - -	mitigated neurotoxicity of a G_4_C_2_ hexanucleotide repeat expansion in the *C9orf72* gene, the most common factor leading to ALS and FTD	[18,150,151]
**spinal muscular atrophy (SMA)**	extensive UPF1 targeting of partially functional mutated SMN1 (survival motor neuron) mRNA with premature stop codon	complete loss of SMN1 leading to haploinsufficiency and neurodegeneration	[152,153]
**epilepsy**	UPF1 upregulation	increased NMD, more frequent seizures and epileptogenesis	[154]
**Viral Infections**			
**Ebola**	UPF1 hijacked by Ebola genome	promotes viral replication	[155]
**HIV**	UPF1 hijacked by HIV genome	increased infectivity crucial for virion assemble	[156]
**RNA and DNA viral infections**	- - -	viral genomes as targets to UPF1-mediated SMD and SRD due to their policistronic organization and high GC content	[20]
**Antiprion Systems**			
**prion infections**	- - -	proposed yeast model of antiprion system depending on Upf1 activity for studying human prion infections	[157]

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
