# Peer review of "UPF1—From mRNA Degradation to Human Disorders"

_cells, 2023, doi:10.3390/cells12030419_

Round 1
Author Response
We would like to thank the reviewer for the appreciation of our work and for useful comments. Below, we provide a point-by-point response to each of the comments raised by the reviewer.
General concept comments
- For comprehension, this reviewer suggests discussing functional roles (section 3) prior to structure (section2). The background provided in this section will be beneficial to understanding the detailed information on structure, particularly on interactions with other proteins, for nonexperts.
Response: We decided to discuss structure prior to function because our main goal was to stress the importance of UPF1 structure and how it results in its functions.
- The authors discuss that the CH-domain has been reported to be inhibitory to UPF1 ATPase and helicase [ref 21], while others have reported no effect [ref 22]. This is presumably due to interactions with different domain and/or accessory factor. The statement “Interestingly, Cleric et al. claimed…..” (line 171) suggests this is controversial. Further elaboration on the potential reasons for discrepancies between these studies would be valuable.
Response: Further elaboration on the potential reasons for discrepancies in studies in the discussion about the CH-domain added in lines 178-179.
- In section 4, spinal muscular dystrophy is mentioned in table but not in text whereas prior diseases are discussed in text but not in table.
Response: As noted by the reviewer, paragraph about SMA added in lines 664-668 and about epilepsy in lines 669-680
- The authors should also include the potential role of UPF1 in epilepsy: Mooney CM, Jimenez-Mateos EM, Engel T, Mooney C, Diviney M, Venø MT, Kjems J, Farrell MA, O'Brien DF, Delanty N, Henshall DC. RNA sequencing of synaptic and cytoplasmic Upf1-bound transcripts supports contribution of nonsense-mediated decay to epileptogenesis. Sci Rep. 2017 Jan 27;7:41517. doi:10.1038/srep41517.
Response: Changes in Table 1 – ATF3 definition deleted; “Neurodegenerative” changed to “Neurological”; information about epilepsy added and according to reviewer’s note we included information about prion infections (It was not in the previous version of the table because there is no described role of UPF1 in prion infections in human per se, and the table depicts role of UPF1 in human diseases. There is only a proposed yeast system to study prion infections and we noted it in added information in the table).
- 1 – it is unclear where the SQ regions (cyan) are located on schematic, perhaps right side of figure is cut-off?
Response: Figure 1 position corrected – the displacement was possibly caused by edditorial changes, in the file supported by us the whole figure was visible.
Specific comments
- Figures are helpful in understanding concepts, but should be referenced earlier in text (e.g. Fig 1 near beginning of section 2; Fig 2 near beginning of section 2.1, etc)
Response: Reference to Figure 1 added in lines 64 and 96; to Figure 2 in line 109.
- For Table, better separation between rows for diseases would facilitate comprehension (ie. aberration, effect), maybe landscape format
Response: we corrected the separation between rows in the table and prepared the table in the landscape version.
- References from line 303-338 are highlighted?
Response: we checked and corrected.
- The text is dense and contains many acronyms. These are not always defined at first reference and/or defined multiple times. Eg. abstract – PTC and NMD not defined (line 12); CH domain (line 65) not defined until following paragraph (line 95); SURF (line 212, 312) and PTC (line 300, 493) defined multiple times; ATR (line 365), G3BP1 (line 387), YTHDF (line701) not defined; etc
Response: PTC and NMD defined in abstract – lines 13, 14; NMD definition deleted in line 33 and repeated NMD substituted with “it”; PTC definition deleted in lines 48-49, 310-311, 509; SMG1 definition corrected from “suppresor of morphogenesis effect in genitalia 1” to “suppresor of morphogenesis in genitalia-1", SMG2 definition deleted in line 80; TRIM25 definition added in line 197; Ebs1 definition added in line 212; “MS” changed to “mass spectrometry” in line 215; BTZ definition added in line 264; Y14 alternative name and its definition added in line 265; MAGOH definition added in line 265; Definition of NMD corrected in lines 293, 306; SLBP definition added and the repeated word “protein” deleted in line 295; G3BP1 definition added and the repeated word “protein” deleted in line 297; SURF definition deleted in line 323; ATR definition added in line 376-377; TLR4 definition added in line 408-409; SMG1C word added in line 424; PVT1 and ATF3 definitions added in line 542; NSCLC abbreviation added in line 564; U2AF1 definition added in line 572; SF3B1 Definition added in lines 572-573; SNHG6 definition added in line 620; GSPT1 definition added in lines 706-707; YTHDF definition added in line 766.
- Multiple spelling (ie. taggressomes line 496) and grammatical errors (ie. missing comas and periods) to address
Response: We checked spelling errors – kDa spelling corrected in line 81; “Clerici” spelling corrected in line 175; “STAU 1proteins” changed to “STAU1 proteins” in line 360; space added in lines 428, 429, 458; “reveled” spelling corrected in line 497, “taggresomes” spelling corrected in line 512; “such a” changed to “such as” in line 555
Grammatical errors corected – “folds” changed to “fold” in line 63; “the” and “a” added in line 83; “the” added in lines 123, 156, 193, 198, 200, 281, 328, 436, 556; “a” added in lines 173, 296, 303, 323, 458, 460, 494, 655; “in agreement to” changed to “in agreement with” in lines 187 and 237; “an” added in lines 199, 220, 251, 355; “seem” changed to “seems” in line 229, “the” deleted in lines 244, 324 (twice), 386, 422, 466; “its” changed to “their” in line 313; “eliminate” changed to “eliminates” in line 335; “influence” changed to “influences” in line 556, “have” changed to “has” in line 595, “have” changed to “has” in line 753
Punctuation mistakes – 22 missing commas added – lines 54, 66, 104, 146, 193, 218, 302, 324, 341, 344, 425, 480, 485, 514, 525 (twice), 595, 708, 710, 711, 744, 746; bracket deleted in lines 74, 524 and 525; two missing periods added in lines 187 and 619; comma deleted in line 335; dash added in “stem-loop” in lines 372, 373 and 383; dash added in “UPF1-dependent” in line 397; dash added in PNRC2 definition in line 432
Stylistic errors – “couldn't” changed to “could not” in line 184; “to the” deleted in line 344; “that” changed to “which” in line 377; “yet” deleted in line 467; “many”changed to “few” in line 756
- Define CH domain (line 65); this is not done until following paragraph (line 95)
Response: we corrected CH-domain defined in lines 66-67, definition deleted in line 152
- PTCs defined in both line 300
Response: we corrected
- Sentence – line 70-75
Response: we changed to “Numerous factors responsible for those processes, e.g., UPF2 and UPF3 (Up-frameshift 2 and 3 nonsense- mediated mRNA decay factors), suppressor of morphogenesis in genita-lia-1 (SMG1) kinase, PP2A (protein phosphatase 2A), SMG5-7 and exon junction complex (EJC) have been identified in humans and characterized for better understanding of UPF1-protein complexes association, remodeling and NMD activation [24,27,31–34].”
- kDa (line 79), Stau 1proteins (line 349)
Response: we corrected: 130kDA changed to 130 kDa, “STAU 1proteins” changed to “STAU1 proteins”
- “Authors suggest” should be replaced by names (line 110, 266)
Response: “Authors” changed to authors’ names in lines: 112 “Cheng et al.”, 276 “Melero et al.”
- 703 residue should denote a.a. (eg. K498, R865; line 123)
Response: Residue description (R703) added in line 126
Additional changes
- Citation linked in one bracket in line 117
- Citation lost during our editorial process added in lines 180-181, 191, 191
- Subsection named UPF1 in anti-prion systems moved after subsection UPF1 in viral subsection from lines 681-702 to lines 718-739, taking into consideration lack of direct role of UPF1 in human prion infections
- Previous references from lines 790 to 1122 deleted and replaced by updated references in lines 1124-1472
Reviewer 2 Report
Comments and suggestions are described in the attached document.

Author Response
First of all, we would like to thank the reviewer for the appreciation of our work and for useful comments. Below, we provide a point-by-point response to each of the comments raised by the reviewer.
While reading this article, I noticed that there were a few instances where additional citations would have been helpful in providing more context and supporting the authors'claims. I understand that space constraints can sometimes make it difficult to include all relevant citations, but I believe that proper referencing is an important aspect of scientific writing.
To be specific I will clarify lines that, when cited, will improve the quality of the
article and/or explain the claims of the text:
- The sentence in lines 263 to 265 explains how further studies elucidated the interaction of SMG1C with RecA-like domains, and independently UPF2 with SMG1, but no citations of those studies have been attached to that sentence.
- Lines 354 to 359 explain histone synthesis and regulation. Although they
cover general aspects of biology a citation would have been appreciated.
- The introduction to Regnase-1-mediated mRNA decay in lines 389 to 393 has the same issue. Although it is a clear and concise introduction there is no citation during the first two sentences.
- Claims in lines 453 to 457 refer to the investigation of NMD inhibition upon viral attack, and how UPF1 targeting is presented in many studies as a future solution. Besides those claims referencing previous investigations and many studies, no references are provided in the sentence.
- The previous error is recurrent in lines 462 to 464 where it is stated that “some research declares that UPF1 has a function in addition to that and is involved in the degradation of peptide”. The vague description of some research in combination with the lack of citations creates the necessity for reformating.
- The aggresome formation and its relationship to the phosphorylation are
described in lines 501 to 509, where one or multiple references should be considered.
- The downregulation of UPF1 in numerous cancers is described in lines 529 to 533. Although Table 1 provides information on UPF1 across diseases, the specific citations to the cancer types referred to in those sentences would have been appreciated.
- Lastly, sentences in lines 692 to 698 conclude in many studies where novel activities and processes depending on UPF1 activity were demonstrated, but no references are provided for those specific lines.
Response: According to reviewer’s comments citations added in lines 276 (interaction of SMG1C with RecA-like domains), 370 (histone synthesis and regulation), 407 (introduction to RMD), 479 (peptide degradation), 518 (aggresome formation), 551 (UPF1 downregulation in cancers), 757 (conclusions), 760 (conclusions)
Due to reviewer’s comment, sentence rephrased to better express author’s intention in lines 470-471 (viral attack)
Citations added in lines not noted by the reviewer: 522, 523, 526
Changes in citations numbers in lines 372, 381, 384, 385, 387, 390, 391, 394, 398, 401, 409, 410, 412, 419, 423, 428, 433, 438, 446, 449, 452, 457, 460, 464, 474, 484, 491, 505, 513, 516, 517, 528, 529, 534, 535, 537, 538, 554, 556, 558, 560, 562, 564, 566, 571, 575, 578, 586, 589, 591, 593, 595, 601, 606, 609, 610, 616, 619, 622, 623, 626, 631, 632, 633, 638, 646, 653, 655, 658, 663, 711, 714, 717, 720, 722, 723, 727, 735, 737, 742, 745, 747, 751, 754, 765, 770, 771, 775, 778